# Yeast Protein Kinase A Isoforms: A Means of Encoding Specificity in the Response to Diverse Stress Conditions?

**DOI:** 10.3390/biom12070958

**Published:** 2022-07-08

**Authors:** Declan R. Creamer, Simon J. Hubbard, Mark P. Ashe, Chris M. Grant

**Affiliations:** 1Division of Molecular and Cellular Function, School of Biological Sciences, Faculty of Biology, Medicine and Health, The University of Manchester, Manchester M13 9PT, UK; declan.creamer@manchester.ac.uk (D.R.C.); mark.p.ashe@manchester.ac.uk (M.P.A.); 2Division of Evolution, Infection and Genomics, School of Biological Sciences, Faculty of Biology, Medicine and Health, The University of Manchester, Manchester M13 9PT, UK; simon.hubbard@manchester.ac.uk

**Keywords:** protein kinase A, isoform specificity, stress response

## Abstract

Eukaryotic cells have developed a complex circuitry of signalling molecules which monitor changes in their intra- and extracellular environments. One of the most widely studied signalling pathways is the highly conserved cyclic AMP (cAMP)/protein kinase A (PKA) pathway, which is a major glucose sensing circuit in the yeast *Saccharomyces cerevisiae.* PKA activity regulates diverse targets in yeast, positively activating the processes that are associated with rapid cell growth (e.g., fermentative metabolism, ribosome biogenesis and cell division) and negatively regulating the processes that are associated with slow growth, such as respiratory growth, carbohydrate storage and entry into stationary phase. As in higher eukaryotes, yeast has evolved complexity at the level of the PKA catalytic subunit, and *Saccharomyces cerevisiae* expresses three isoforms, denoted Tpk1-3. Despite evidence for isoform differences in multiple biological processes, the molecular basis of PKA signalling specificity remains poorly defined, and many studies continue to assume redundancy with regards to PKA-mediated regulation. PKA has canonically been shown to play a key role in fine-tuning the cellular response to diverse stressors; however, recent studies have now begun to interrogate the requirement for individual PKA catalytic isoforms in coordinating distinct steps in stress response pathways. In this review, we discuss the known non-redundant functions of the Tpk catalytic subunits and the evolving picture of how these isoforms establish specificity in the response to different stress conditions.

## 1. Introduction

The ability of cells to sense and rapidly respond to the challenges that are posed by their external environment is vital for their survival. Through fitness that is conferred by natural selection, organisms have evolved mechanisms to sense changes in their environment, such as fluctuations in temperature and nutrient availability, and mount appropriate stress responses to maintain cellular homeostasis [1,2]. Signal transduction pathways play a major role in responding to stresses by controlling enzyme cascades that ultimately promote remodelling of the transcriptome and proteome in multi-stress and stress-specific manners [1,2,3,4]. These signalling cascades involve multiple steps, including the production of second messenger molecules e.g., cyclic nucleotides, the subsequent activation of kinases which phosphorylate downstream substrates, and the fine-tuning of these signals by protein phosphatases.

One key signalling pathway that is conserved among eukaryotes is the cyclic AMP (cAMP)/protein kinase A (PKA) pathway, which plays a major role in the control of cell growth and proliferation. PKA was first identified over 50 years ago as a cAMP-dependent heterotetrameric–holoenzyme complex, consisting of two catalytic (C) subunits and a regulatory (R) subunit dimer (R_2_C_2_) (reviewed in [5,6]). In *S. cerevisiae*, the C subunits are encoded by the genes *TPK1, TPK2* and *TPK3,* promoting the expression of three independent isoforms, while the R subunit is encoded by a single gene, *BCY1* [7]. In comparison, higher eukaryotes have evolved further complexity at the level of the PKA subunits, with several genes and alternative splicing events giving rise to multiple subunit variants [8,9]. *S. cerevisiae* is one of the few known fungal species to express more than two PKA catalytic subunits however [10], and has therefore been used as a model eukaryote to study the role of specificity in PKA signalling [11].

PKA activity is essential in *S. cerevisiae* since at least one catalytic isoform is required for viability [12]. The activity of these yeast Tpk isoforms is multifaceted and controls many developmental and adaptive processes, including cellular growth, carbohydrate metabolism, the cell cycle and the general response to stress [1,13,14,15]. Tpk1-3 are believed to be redundant in supporting many of these functions, although an increasing number of studies have highlighted various processes for which isoform specificity is a key component in achieving targeted signalling. Notably, isoform specificity has been shown as important in the response to different environmental stresses. This gives rise to key questions concerning how signalling specificity is achieved under different stress conditions if different PKA kinase isoforms are important in dictating the varied stress responses [2]. In this review, we focus on the identified functions of the individual Tpk isoforms as they exist in *S. cerevisiae.* We consider the strategies that are employed by yeast cells to direct and compartmentalise PKA signalling and discuss recent research that is aimed at unravelling the molecular mechanisms which encode Tpk specificity in the response to diverse stress conditions. 

## 2. Glucose-Induced cAMP Signalling

The PKA pathway in *S. cerevisiae* is a key mechanism by which cells detect nutrients such as glucose, facilitated by two systems functioning in concert to stimulate adenylate cyclase (Cyr1) activity (Figure 1). The transmembrane G-protein coupled receptor (GPCR) Gpr1 binds glucose as an extracellular ligand, and, acting via the G-alpha subunit Gpa2, signals to Cyr1, which triggers production of the second messenger cAMP from ATP [15,16,17,18,19,20]. However, for Cyr1 to be sensitive to the increase in glucose concentration that is signalled by Gpr1, it must first be potentiated by Ras1 and Ras2 [19]. Ras proteins are a widely studied family of small GTPases conserved in lower and higher eukaryotes and play crucial roles in multiple signalling pathways involved in cell growth and proliferation, including the cAMP/PKA and mitogen-activated protein (MAP) kinase pathways [21,22,23]. The Ras system also responds to glucose, since intracellular glucose that is metabolised by the glycolysis pathway activates a guanine nucleotide exchange factor (GEF), Cdc25, that acts on Ras1/2 [19,24,25]. GTP bound Ras1/2, anchored predominantly at the plasma membrane and organellar membranes, are activated, allowing full stimulation of Cyr1 [26,27]. Downregulation of the Ras proteins is controlled by the GTPase activating proteins (GAPs) Ira1/2, which stimulate the intrinsic GTPase activity of Ras1/2 to hydrolyse GTP to GDP, thus returning them to an inactive state [28,29]. Therefore, this complex circuitry leads to transient changes in cAMP levels, resulting in the modulation of PKA activity and the fine control of a broad range of cellular processes.

## 3. The Mechanics of PKA Enzymatic Control by cAMP and Regulatory Subunits

Constitutively active PKA activity is detrimental to yeast cells. For example, deletion of the *BCY1* gene, resulting in a loss of PKA regulation, causes heat sensitivity, defects in sporulation and sensitivity to nutrient starvation [7]. In fact, sensitivity to multiple stresses including heat and oxidative stress broadly correlates with cellular cAMP levels, emphasising the importance of the cAMP/PKA pathway in stress tolerance [30]. 

Not surprisingly therefore, cAMP levels are themselves tightly regulated and can be decreased by the activities of cAMP phosphodiesterases (PDEs) which hydrolyse cAMP to AMP. *S. cerevisiae* has two cAMP phosphodiesterases, high affinity (Pde2) and low affinity (Pde1) enzymes [31] (Figure 1). Pde2 is thought to regulate basal cAMP levels, whereas Pde1 functions after PKA activation to dampen cAMP signalling in a negative feedback signalling loop [32]. In mammalian cells, it is well established that phosphodiesterase enzymes are compartmentalised at specific sites, which spatially constrain cAMP signalling and direct local PKA activation [33]. Evidence in yeast has shown that Pde2 localisation is altered in a PKA-dependent manner [34], suggesting that its cellular location might be important in regulating the feedback inhibition of PKA activity. However, it is currently unknown whether PDEs in yeast regulate cAMP levels and PKA activity via compartmentalisation, similar to those in mammals.

The PKA regulatory subunit (Bcy1) is the only cAMP receptor molecule that is expressed in yeast. In the absence of cAMP, Bcy1 interacts with the catalytic subunits, Tpk1-3, in an inactive R_2_C_2_ complex (reviewed in [11,16,31]). After a rise in intracellular cAMP levels, for instance after glucose detection, cAMP binds to two cyclic nucleotide binding (CNB) domains that are present on each Bcy1 monomer [6,11]. This induces allosteric changes in the holoenzyme structure, which have been proposed to result in the dissociation of the R_2_C_2_ complex and release of the catalytic subunits. 

Recent studies in mammalian systems have suggested that full dissociation of the R subunits from the catalytic C subunits is not essential for PKA holoenzyme activity that is consistent with the more localised targeting of PKA substrates [35,36]. For example, the simultaneous addition of cAMP and substrate has been shown to differentially affect the extent of type I and type II PKA holoenzyme dissociation using in vitro purified proteins [37]. However, this ‘loosening’ model proposed by the Scott laboratory has been challenged, as it was later demonstrated that although the majority of C subunits are released from R subunits upon cAMP binding, the molar excess of R over C subunits likely limits their diffusion and promotes efficient recapture [38]. Interestingly, some studies have also shown a role for the PKA substrates themselves in Tpk enzyme activation [39]. Consequently, the mode of PKA activation in yeast remains poorly understood, and the classical model that the full dissociation of the R from the C subunits occurs in response to cAMP binding may represent an oversimplification of the activation process. 

Tpk isoenzymes phosphorylate the same basophilic recognition sequence (RRXS*/T* motif) but phosphoproteomics has revealed that the isoforms target different substrates, raising the question as to how substrate specificity arises [40]. In mammalian cells, substrate specificity can be driven by scaffold proteins called A-kinase anchoring proteins (AKAPs), which are responsible for determining the subcellular localisation of the holoenzyme and subsequent access to its activators and substrates [41]. Unlike the single yeast R protein, there are four mammalian R protein isoforms which display differential affinities with AKAPs. A structural analysis of Bcy1 revealed that the N-terminal region that is responsible for dimerization and for docking to AKAPs (D/D domain) in higher eukaryotic R subunits is not well conserved in yeast Bcy1 [42]. The AKAP regulation of yeast PKA activity therefore remains ill-defined, although a screen for Bcy1-interacting proteins identified potential candidate proteins including the Hsp60 mitochondrial chaperone that might function in the localisation of Tpk1 to mitochondria [43]. 

Bcy1 is normally localised to the nucleus in glucose-grown cells but displays a broader nucleocytoplasmic localisation in carbon source-derepressed cells [44]. This carbon source-dependent localisation of Bcy1 depends on the phosphorylation of Bcy1 via the Yak1 kinase, although it is unclear whether the nucleocytoplasmic distribution of Bcy1 affects PKA activity in an isoform-specific manner. It is possible that moderating Bcy1 localisation and hence binding to the catalytic subunits provides a mechanism to globally alter PKA activity. Some preliminary data suggested that the interaction between Tpk1/2 and Bcy1 is stronger than that of Tpk3, suggesting differential regulation of the C subunits by Bcy1 [45]. However, little is known regarding individual catalytic subunit affinities with the single Bcy1 regulatory subunit and how this relates to their intracellular distribution during different growth conditions. 

## 4. Unravelling Tpk Isoform Specificity 

The three PKA catalytic subunits in *S. cerevisiae* were originally isolated using genetic techniques and shown to be homologous to the bovine cAMP-dependent protein kinase [12]. The deletion of all subunit genes rendered cells inviable, but the retention of any single *TPK* gene was sufficient to rescue this lethality, suggesting redundancy among the three subunits for essential functions [12]. Tpk1-3 are similar in terms of protein sequence and domain structure, possessing a conserved C-terminal kinase domain (~300 residues) and relatively short N- and C-terminal extensions (~40 residues) (Figure 2) [12,46]. They also share a 75% sequence similarity across the kinase domain overall, with Tpk1 and Tpk3 exhibiting a slightly higher sequence identity (88%) due to their paralogous nature [12,47]. The highest variability is present in the N-terminal domain (NTD) of each isoform, with this being consistent with the observation that the catalytic subunits of unicellular eukaryotes show considerable variation in their N-termini relative to one another [48]. Such variability is particularly evident in the NTD of Tpk2. This possesses glutamine-rich regions that are predicted to form a prion-like domain (PrLD), which are absent from Tpk1/3 [49].

Evidence that has been gathered over the last three decades has shown that catalytic subunit isoforms are not equivalent in all respects and exhibit specificity in a diverse range of processes such as filamentation, iron regulation and mitochondrial biogenesis [46,50,51,52]. It is thought that distinct PKA catalytic subunit roles might be defined by their divergent N- and C-terminal extensions, which likely facilitate interactions with different proteins and phosphorylation targets [9,46]. The concept of C-kinase anchoring proteins (CKAPs), which encompass all proteins that physically bind PKA catalytic subunits, has been suggested in mammalian cells as a means of localising C-subunits to specific subcellular compartments, thus creating ‘pools’ of isoform-specific phosphorylation [9]. For example, studies in renal cells that mutated in different PKA catalytic subunits have revealed large-scale phosphoproteomic differences, in spite of the isoforms phosphorylating nearly identical consensus motifs [53]. These differing phosphorylation patterns add weight to the idea that differences in substrate phosphorylation are mediated by differences in protein interactions. Although the regulation of mammalian PKA is more complex due to the existence of multiple catalytic subunit splice variants and regulatory subunit isoforms, these principles could be extended to other eukaryotes, including S. cerevisiae, where PKA subunits are observed at distinct subcellular locations that are dependent on nutritional status and stress exposure [54,55]. Identifying the interacting partners of Tpk1-3 could therefore serve as a way of uncovering the potential spatial regulation of the different isoforms. Alterations in the protein–protein interaction networks associated with the response to cellular stress are also likely key to the changing localisation that is observed for the Tpk isoforms during stress conditions, as discussed later. 

### 4.1. PKA-Specific Signalling Mediates Pseudohyphal Growth

Early studies on filamentous growth in *S. cerevisiae* highlighted a role for Ras2 activation in stimulating the pseudohyphal response [57,58]. The *RAS2*^val19^ dominant active mutant was shown to induce filamentation via both the mitogen-activated protein kinase (MAPK) and cAMP/PKA pathways; however, it has become clear that they contribute differently to filamentous growth activation [46,59,60]. PKA signalling specificity was first described by Robertson and Fink (1998), who observed differential roles for the Tpk isoforms in supporting filamentous growth. The capacity to grow in a filamentous form is an important fungal developmental response to stress and has been suggested to enable yeast such as *S. cerevisiae* to forage for nutrients and/or escape accumulating toxins. The formation of chains of elongated cells that remain connected to one another and have the capacity to invade solid growth media involves alterations in the cell cycle, flocculence and budding pattern. Studies on the role of PKA showed that *TPK2* deletion ameliorated the filamentous growth response, while *tpk1* mutant cells showed a phenotype that was indistinguishable from the wild-type [46]. In contrast, the loss of *TPK3* induced hyperactive filamentous growth, highlighting opposing roles for the Tpk2 and Tpk3 isoforms [46,61]. 

Mechanistically, the Tpk2-dependent regulation of filamentous growth is thought to involve the phosphorylation and targeting of two transcription factors, Sfl1 and Flo8. These factors act antagonistically in the transcriptional regulation of a key cell wall, flocculin Flo11, and *FLO11* transcription represents a critical determinant of filamentous growth [46,62,63]. More specifically, Tpk2 inhibits the Sfl1 repressor, while acting as a positive regulator of Flo8 [61]. Interestingly, many standard laboratory yeast strains have lost their capacity to undergo filamentous growth through mutations in the Flo8 transcription factor [64] and strains that are mutant in *FLO11* are deficient in invasion, filamentation and biofilm formation. Therefore, the overall Tpk2-dependent phosphorylation of key transcription factors is postulated to activate *FLO11* transcription to promote the switch to a filamentous morphology [61]. 

Although earlier research considered Tpk1 to have no direct role in filamentous growth [46], more recent studies have identified Tpk1 as playing a regulatory role via the dual-specificity tyrosine kinase Yak1 [60,65,66]. Yak1 was found to be part of a regulatory cascade which induced *FLO11* transcription through activation of the transcription factors Phd1 and Sok2 [60,66]. Yak1 phosphorylation by Tpk1 abolishes this activation to limit *FLO11* induction [60,65]. These findings highlight how Tpk isoforms can have more discrete functions in cellular processes by acting through networks of connected signalling cascades. 

### 4.2. Requirement for PKA Activity in Mitochondrial Function and Biogenesis

Early links with mitochondrial function came from a key study that identified functional signatures for each of the three catalytic subunits via a genome-wide microarray analysis of yeast mutants lacking the individual Tpk isoforms [50]. Using this transcriptomic analysis, loss of *TPK1* was found to decrease the expression of the genes *BAT1* and *ILV5*, both of which are involved in branched chain amino acid (BCAA) biosynthesis [67]. Removal of *ILV5* causes the deletion of large portions of mitochondrial DNA (mtDNA) [68], while Bat1 also functions in mitochondrial iron homeostasis [67]. This study therefore suggests that Tpk1 is required for maintaining the integrity of the mitochondrial genome and in the derepression of BCAA biosynthesis. In this expression analysis, Tpk2 was also suggested to play a role in iron regulation, as genes that were implicated in high affinity iron uptake such as *FTR1* were upregulated in a *tpk2* mutant [50]. Ftr1, which encodes a plasma membrane-localised high affinity iron transporter, has since been shown to activate the PKA pathway upon addition of iron to iron-starved cells [69]. This evidence suggests that once PKA is active and the requirement for respiration and thus iron is reduced, Tpk2 could operate via a negative feedback loop to downregulate transceptor gene expression and prevent the uptake of additional iron. 

Curiously, this array analysis found that no transcriptional changes were attributed to the loss of *TPK3*, aside from those that were described for pseudohyphal growth [50]. Tpk3 is considered to have the lowest catalytic activity of the three isoforms due to poor expression of the *TPK3* gene [70]. However, later studies confirmed that Tpk3 has non-redundant roles in both respiration and mitochondrial regulation [51,52,71]. Mitochondrial PKA substrates have been described, including at least eight proteins which are differentially phosphorylated by Tpk1-3, albeit with significant overlap [72]. Experiments which first suggested that Tpk3 functions directly in mitochondrial regulation showed that *tpk3* mutant cells have significantly reduced respiratory rates and possess modified mitochondria [51]. Indeed the mitochondria contain reduced levels of key respiratory enzymes such as cytochrome *c*, which has previously been proposed as a PKA target [51,73]. In addition to controlling enzyme content, Tpk3 was suggested to regulate elements of the mitochondrial biogenesis machinery, as in its absence, the levels of reactive oxygen species (ROS) that were generated by mitochondria were greatly increased [71]. Hyperactivation of the PKA pathway via *PDE2* deletion has also been shown to produce effects on mitochondria. Mitochondria are irregularly shaped and genes that are involved in the electron transport chain and ROS detoxification are down regulated. These unusual mitochondrial phenotypes and transcriptional effects are dependent on the expression of *TPK3* [52]. Taken together, these studies highlight the key role that is played by Tpk3 in regulating mitochondrial biogenesis and function. 

## 5. Role of the cAMP/PKA Pathway in Stress Tolerance

The ability of organisms to adapt to and navigate the challenges that are posed by their external environments is vital for survival. This is of particular importance for non-motile, unicellular eukaryotes such as yeast, which are often exposed to fluctuations in nutrient availability, temperature, levels of ROS and pH, etc. One such stress response that is activated by multiple diverse stressors, including glucose starvation, heat and oxidative stress, is the environmental (or general) stress response (ESR) [1,74,75,76,77,78] (Figure 3). The ESR is the best-characterised example of a stress response programme that is regulated by PKA, as the activity of many factors which stimulate the transcription of ESR-responsive genes are regulated by PKA-mediated phosphorylation [76,79,80]. Broadly speaking, PKA is rapidly inactivated in response to various stresses, which downregulate processes that would otherwise be deleterious for cellular survival during exposure to environmental insults [81]. These include transcription, translation, glycogen synthesis, cell cycle progression and cell growth. The removal of PKA activity simultaneously derepresses other key signalling molecules including selected kinases and transcription factors (TFs), which are then free to induce a multitude of genes that promote stress resistance and survival [16]. 

A large proportion of genes that are induced by the ESR contain a stress response element (STRE) in their promoter regions which acts as a binding site for the homologous zinc-finger TFs, Msn2 and Msn4 [16,76,82]. The STRE was originally identified as part of a Hsf1-independent transcription induction mechanism for the heat-induced genes, *DDR2* and *CTT1* [83,84]. Msn2/4 are among the most significant downstream targets of PKA, as their phosphorylation at functionally important motifs affords PKA control over the expression of a broad range of genes [1,80,85]. Additionally, the loss of *MSN2* and *MSN4* was found to rescue the lethality of a triple *tpk* mutant, confirming the key role that these transcription factors play in moderating the global effects of cAMP/PKA on cell growth [74,80]. A subsequent study revealed that Tpk1 and Tpk3 play more predominant roles in suppressing Msn2/4 activity, highlighting potential isoform specificity in controlling the ESR (Figure 3) [86].

Both Msn2 and Msn4 exhibit oscillatory behaviour and translocate between the nucleus and cytoplasm in a PKA- and karyopherin-dependent manner [76,80,87,88]. Under normal growth conditions, high PKA activity constrains the dynamics of Msn2/4 via increased phosphorylation of a conserved nuclear localisation signal (NLS), thus confining the majority of Msn2/4 molecules to the cytoplasm [88,89]. Conversely, after stress and the subsequent lowering of PKA activity, Msn2/4 become dephosphorylated and hence bind to nuclear import factors, leading to their accumulation in the nucleus [88]. In the case of Msn2, Kap121 and Kap123 are involved in its nuclear targeting [90], while its export is regulated by Msn5 via binding to its nuclear export sequence (NES) (Figure 3) [91]. The Msn2 NES is phosphorylated by nuclear PKA which promotes binding to Msn5 and increases the rate of its exit from the nucleus during non-stress conditions (Figure 3) [83,88]. This is abrogated under stress conditions, which retains Msn2 in the nucleus and allows binding to STRE-containing genes [87]. The retention of Msn2 in the nucleus can be detrimental however, and chronic nuclear accumulation results in degradation of the protein to avert a prolonged ESR [87]. The relative contributions of nuclear and cytosolic pools of Tpk proteins in regulating Msn2/4 oscillations are not well understood. It is therefore plausible that each isoform exhibits specificity for phosphorylating the NLS vs. NES, especially given that Tpk1 and Tpk3 are proposed to suppress Msn2, whereas Tpk2 is a partial activator [86]. 

## 6. Control of the Heat Shock Response by the cAMP/PKA Pathway

Fluctuations in temperature represent one of the most fundamental stresses that are experienced by yeast cells. It is widely accepted that heat stress can cause proteins to misfold, lose their native structures or adopt non-native states that are prone to aggregation. Stress-induced protein misfolding and aggregation can be cytotoxic, although this toxicity can be mitigated by the activity of molecular chaperones including members of the heat shock protein family [92,93]. Newly synthesised polypeptides are particularly susceptible to misfolding and so can require extensive remodelling via chaperones to assume their native states [92]. 

At suboptimal temperatures, yeast cells activate a stress response pathway that is known as the heat shock response (HSR) [2]. This transcriptional programme is triggered in response to diverse stress conditions including heat, mistranslation, osmotic and oxidative stresses, and is required to transiently regulate the expression of proteostatic genes such as those encoding chaperones [2,94]. The HSR is orchestrated in eukaryotes by the evolutionarily conserved transcription factor, heat shock factor 1 (Hsf1) [95]. Hsf1 is essential in yeast, suggesting that it is also required during normal, non-stress conditions [96]. It has been shown to function downstream of the cAMP/PKA pathway, and like the ESR it is negatively regulated by PKA activity [97]. Not surprisingly therefore, differences are observed in the cAMP/PKA targeted phosphoproteome upon heat shock [98]. These phosphoproteomic changes include the rapid dephosphorylation of the PKA phosphorylation sites within the Msn2/4 nuclear localization signal (NLS), as well as concomitant changes in the phosphorylation of other PKA substrate proteins. The Yak1 and Rim15 protein kinases phosphorylate and activate Hsf1 activity and are themselves inhibited by PKA [99]. Yak1 and Rim15 also phosphorylate Msn2, suggesting an overlapping mechanism moderating the ESR and HSR under the control of the cAMP/PKA pathway. Again, Tpk isoforms are not redundant in their capacity to moderate Hsf1 activity, since Tpk1 and Tpk3 play predominant roles in repressing the HSR [100]. The negative regulation of Hsf1, Yak1 and Rim15 by PKA is consistent with PKA activity being downregulated in response to heat stress, resulting in the dephosphorylation of proteins such as Msn2/4 and thus promoting transcriptional changes that lead to adaptation. 

As high PKA activity is linked to lower stress tolerance, it seems plausible that regulation of PKA subunits at the transcriptional level might represent a mechanism to fine-tune PKA activity in response to stress. Some work has been carried out to study the regulation of *TPK1-3* and *BCY1* during heat shock and has identified that the *TPK1* isoform is specifically upregulated, while *TPK2* and *TPK3* show no such increase in expression [101,102,103]. Curiously, a recent study of Tpk1 regulation during recurrent heat shock has shown that although *TPK1* mRNA becomes more abundant, it is translationally inhibited and stored in mRNA granules until the stress is removed (Figure 4) [103]. During a recovery period, Tpk1 protein levels increase, suggesting that the transcriptional induction of *TPK1* during the original stress serves as an adaptive measure to facilitate increased Tpk1-dependent phosphorylation upon recovery [103]. This emphasises an important role for the Tpk1 isoform and highlights how signalling specificity is required for stress adaptation such that cells are protected from future stressful events. 

It is now well documented that heat shock causes translational inhibition and the formation of biomolecular condensates, termed mRNA processing bodies (PBs) and stress granules (SGs) [104,105,106]. These condensates represent specialised microenvironments containing mRNAs, components of the translational machinery and other proteins that are important for coordinating RNA fate. Temperature fluctuations have been found to cause isoform-specific alterations in Tpk catalytic subunit cellular localisation and their association with stress-induced condensates (Figure 4). For example, in response to severe heat stress, Tpk2 and Tpk3 proteins are targeted to PBs and SGs, while Tpk1 remains within the nucleus [107]. For Tpk2, the re-localisation is dependent on its kinase activity, whereas a catalytically inactive form of Tpk3 maintains its capacity to associate with granules [107]. Yeast strains that were deleted for *TPK2* or *TPK3* also exhibited differences in the expression of multiple mRNAs during heat stress, with Tpk2 seemingly responsible for the resumption of translation initiation after stress and Tpk3 promoting translational repression [107]. Further study has since revealed that a prion-like domain predicted in the N-terminus of Tpk2 is required for its re-localisation to PBs upon heat stress [49]. Tpk3 does not carry this N-terminal PrLD and so the mechanism by which this isoform associates with granules is clearly distinct to that of Tpk2 (Figure 4). This highlights how specific PKA structural domains are involved in condensate formation and how, as in mammalian systems, protein localisation likely drives targeted phosphorylation to modulate key processes such as translation and mRNA fate. 

It is known that tolerance to other HSR-activating stress conditions which cause protein misfolding, such as mistranslation caused by the proline analogue L-azetidine-2-carboxylic acid (AZC), also require moderation of the cAMP/PKA pathway [108]. The downregulation of PKA activity in response to AZC stress is found to be dependent on the oxidation of the Bcy1 regulatory subunit that is mediated by the Tsa1 peroxiredoxin (Prx), which suppresses cAMP binding to Bcy1 and hence limits PKA activation [108]. It is interesting to note that despite the significant role of Tsa1 in the response to AZC stress, Bcy1 oxidation by Tsa1 was not observed in response to oxidant treatment [108]. This Tsa1 mechanism for preventing PKA activation was suggested as a global mechanism acting at the level of the holoenzyme; therefore, any Tpk isoform specificity in the response would appear to be unlikely. 

## 7. Control of the Oxidative Stress Response by the cAMP/PKA Pathway

Crosstalk between oxidant dependent signalling and the PKA pathway is not limited to the effects of Tsa1 on Bcy1. The cAMP/PKA pathway has long been known to be required for the response to oxidative stress. Cells are continually exposed to ROS originating largely from mitochondrial metabolism. In excessive amounts, ROS can evade the coping defence systems that are evolved by cells, resulting in oxidative stress and damage to macromolecules, such as nucleic acids, proteins and lipids [2,109]. Part of the role of the cAMP/PKA pathway stems from its effects on the ESR via the Msn2/4 transcription factors. Additionally, the response to H_2_O_2_ stress is highly dependent on the Yap1 and Skn7 transcription factors, which mediate the major transcriptional response to oxidative stress in yeast [77,110]. Similar to the Msn2/4 mechanism, the Yap1/Skn7 transcription factors are negatively controlled via the cAMP/PKA pathway [111]. While work has been carried out on the relationship between general PKA signalling and hydrogen peroxide stress, little consideration has been given to the roles of individual Tpk isoforms in response to oxidative insults. 

Distinct from changes in transcription, resistance to H_2_O_2_ stress has been attributed to the process of caloric restriction (CR). CR is a mechanism via which organisms from bacteria to mammals can prolong their lifespans, and in yeast, requires the downregulation of signalling pathways such as the cAMP/PKA pathway [112]. The downregulation of PKA in response to CR requires Tsa1, a key Prx with dual functionality as an antioxidant and molecular chaperone [113]. Tsa1 facilitates the reduction in harmful peroxide species but becomes hyperoxidised and retained in an inactive state when it is overwhelmed by ROS. This hyperoxidation is relieved by the activity of sulfiredoxin (Srx1) which reduces oxidised Cys residues in an ATP-dependent reaction [114]. In response to H_2_O_2_ or CR, low PKA activity stimulates the translational induction of Srx1, which increases oxidant tolerance and extends longevity [113]. As ROS production increases with age, the effects that are elicited by PKA downregulation are viewed as an intervention to mitigate oxidative stress.

Somewhat surprisingly, Msn2 activity can be regulated in response to blue light exposure in yeast in a mechanism that is mediated by hydrogen peroxide generated from a peroxisomal oxidase, and sensed by the Tsa1 peroxiredoxin [115]. Tsa1 was found to slow the rate at which Tpk isoforms exited the nucleus as a means of controlling Msn2 localisation (Figure 4). As mentioned previously, the exclusion of Msn2 from the nucleus reduces the expression of stress response genes and is associated with normal growth. Upon H_2_O_2_ treatment, Tpk1 and Tpk2 were found to rapidly exit the nucleus and form cytoplasmic foci, while Tpk3 and Bcy1 remained in the nucleus [115]. This indicates that Tsa1 only partially inhibits PKA, but that this is sufficient to allow nucleocytoplasmic oscillations of Msn2 (Figure 4). Moreover, it suggests that the Tsa1-mediated repression of PKA may act on Tpk1 and Tpk2 but not Tpk3, as Tpk3 localisation is unaffected by Tsa1 expression. 

A more recent study investigating the relationship between PKA signalling and the response to hydrogen peroxide stress identified oxidative post-translational modifications (PTMs) in the Tpk1 catalytic subunit that require Tsa1 activity [116]. A model was proposed where the Tpk1 isoform was repressed via the sulfenylation of Cys243 and the glutathionylation of Cys195/243 when exposed to a bolus of hydrogen peroxide (Figure 4). Cys243 glutathionylation correlates with the increased dephosphorylation of Thr241 in the Tpk1 activation loop, which, when phosphorylated, prevents the autophosphorylation of Ser179 and PKA activation [116,117]. A reduction in Thr241 phosphorylation has been linked to a weakened Tpk1/Bcy1 interaction, which presumably increases the number of active Tpk1 subunits and thus PKA activity [117]. Upon H_2_O_2_ addition, Cys243 sulfenylation increases and there is an increase in the dephosphorylated state of Thr241, which is proposed to destabilise the activation loop and thus inhibit PKA activity. However, it is unknown how the interplay between these PTMs affects the stability of Tpk1-containing PKA complexes and whether these oxidative modifications exclusively inhibit Tpk1 subunits in a monomeric or Bcy1-bound state. Since Tpk1 Cys243 is conserved in Tpk2 and Tpk3 (Figure 2), there remains the strong possibility that Tpk2/3 may also be modified and regulated similarly to Tpk1 upon H_2_O_2_ stress, though this has not been explored. This is an interesting prospect, given that the redox modulation of cysteine residues under oxidative stress conditions appears to be evolutionary conserved in diverse eukaryotic protein kinases [118]. The presence of two additional cysteine residues in the N-terminal domain of Tpk3, which are not conserved in Tpk1/2, may also represent an additional means of regulating PKA during oxidising conditions (Figure 2). 

## 8. Conclusions and Future Perspectives

In many eukaryotes, PKA activity is a composite of differentially regulated catalytic subunits and, although redundant for some functions, it is now clear that isoform specific activity is deployed for different biological processes. According to early data, yeast PKA activity was thought to be globally downregulated in response to a broad array of stress conditions to activate the environmental stress response [1,79,87,90,119]. However, more recent observations that are discussed in this review have challenged this concept and highlighted that PKA activity should no longer be considered as a single entity which is inhibited uniformly. 

Studies in yeast and higher eukaryotes have shown that PKA is subject to several levels of regulation which modulate C subunit activity, including inhibition by the R subunit, phase separation, differential localisation and feedback control. As all PKA C subunits phosphorylate the RRXS*/T* consensus motif, it is likely that kinase-interacting proteins are key to directing substrate phosphorylation by anchoring the subunits to distinct subcellular sites. AKAPs are well characterised in mammalian cells; however, homologues in yeast are yet to be identified. CKAPs, which comprise all the proteins that interact with the C subunits, are therefore a likely determinant in targeting PKA isoforms to specific substrates. In response to stress, PKA has been shown to localise in an isoform specific manner to different cellular compartments, including stress-induced bodies such as PBs and SGs. It is likely that this re-localisation changes the targets of the Tpk isoforms, bringing them near to substrates that are important for stress tolerance. Stressors have also been shown to regulate PKA activity through post-translational modification of the C or R subunits, either directly inhibiting the Tpk subunits or affecting the stability of the holoenzyme complex. Alternatively, it is possible that phosphatases may regulate localised Tpk activity by compartmentalising PKA-dependent phosphorylation, as has been shown in mammalian cells [120,121]. Further careful study is therefore required to understand the heterogeneity of PKA signalling in terms of mapping isoform interaction networks and how the different holoenzyme subunits are regulated during changing cellular conditions. As the kinase activity of certain isoforms is required over others in response to distinct stresses, it will be interesting to dissect the mechanisms which modulate PKA isoforms in response to different conditions and how PTMs may contribute to this. Exciting new advances in the field, such as fluorescence resonance energy transfer (FRET) reporters to monitor cAMP and PKA activity, will also be powerful tools to study isoform specific signalling and its localisation in vivo [122]. A greater appreciation and understanding of the specific roles of kinase isoforms will hopefully facilitate studies to unravel the complexity of the kinome for the isoform specific branches of PKA signalling. 

## Figures and Tables

**Figure 1 biomolecules-12-00958-f001:**
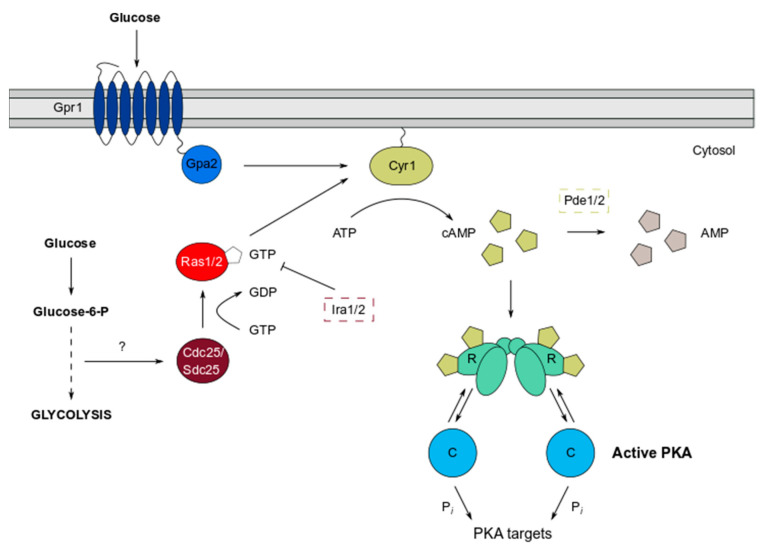
Overview of the cAMP/PKA pathway in *S. cerevisiae.* Pathway activation occurs in response to glucose via the GPCR (Gpr1) and Ras-based systems. These systems collectively stimulate the adenylate cyclase Cyr1 which increases production of cAMP from ATP. PKA is the only known cAMP receptor in yeast, and upon binding the PKA regulatory subunit, cAMP promotes allosteric changes, which liberate the catalytic subunits from the PKA tetramer. This dissociation activates the C subunit kinase activity and enables phosphorylation of PKA targets.

**Figure 2 biomolecules-12-00958-f002:**
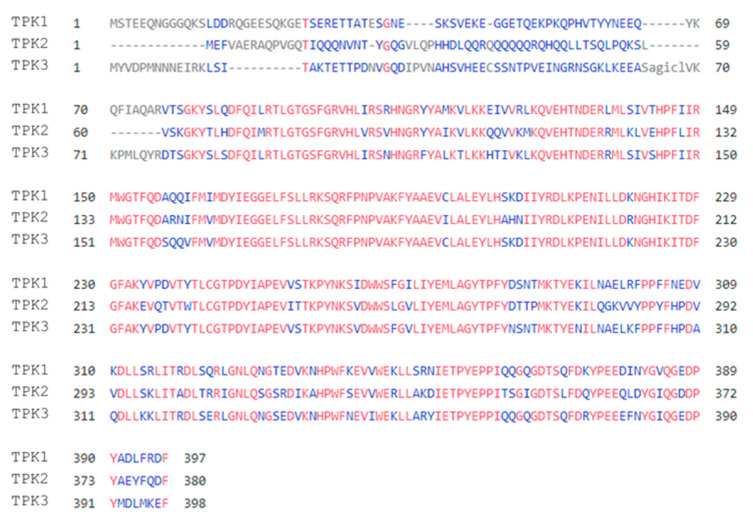
Analysis of the three *S. cerevisiae* PKA catalytic isoform protein sequences. Tpk1-3 sequences (UniProt IDs: P06244, P06245 and P05986, respectively) were aligned and analysed using COBALT [56]. Areas are coloured according to identity between sequences, red showing conservation between the aligned residues and blue showing identity between two Tpk sequences. Low conservation is seen in the N-terminal regions of the three catalytic subunits, with high conservation across the kinase and C-terminal domains.

**Figure 3 biomolecules-12-00958-f003:**
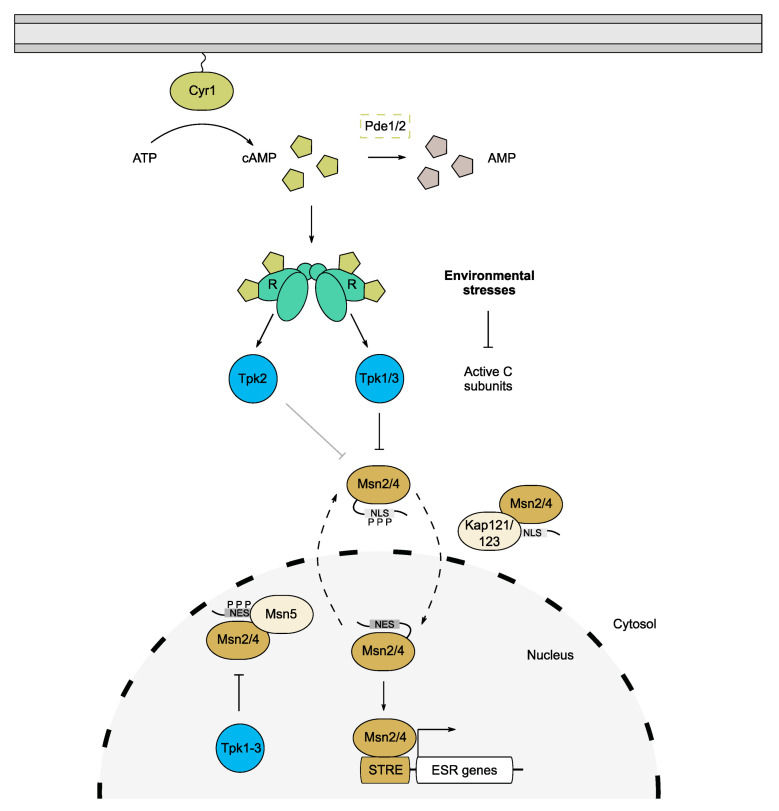
PKA control of the environmental stress response (ESR). Environmental stresses including heat shock, oxidative stresses, osmotic stresses and nutrient starvation constitute a constant challenge for yeast cells. The ESR is mediated by the Msn2/4 transcription factors which are regulated by PKA activity. PKA, predominantly Tpk1/3, phosphorylates the NLS of Msn2/4, preventing transport into the nucleus under normal conditions. Reduction in PKA during stress permits Msn2/4 shuttling and abrogates their association with the export factor Msn5 in the nucleus. The TFs are then free to bind stress response elements (STRE) and promote expression of ESR genes required for stress tolerance.

**Figure 4 biomolecules-12-00958-f004:**
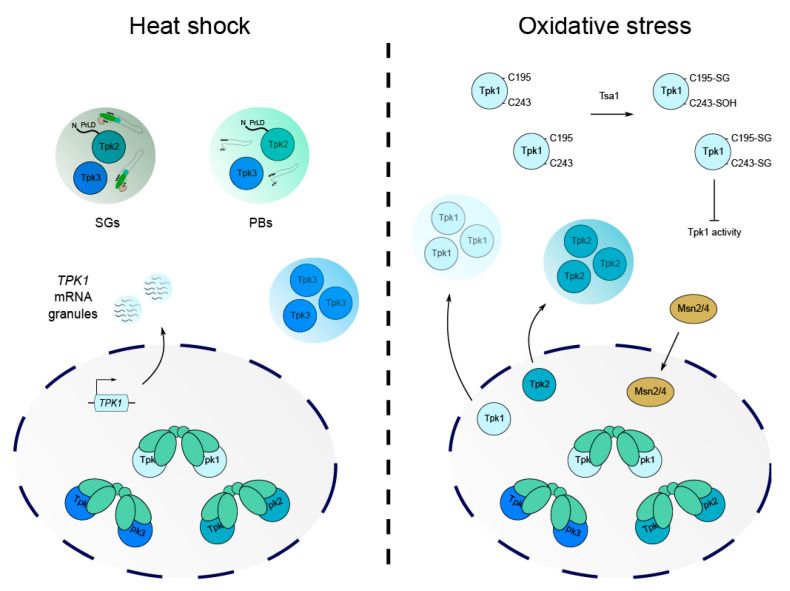
The effects of heat shock and oxidative stress on PKA isoforms. The responses to heat shock and hydrogen peroxide stress both necessitate alterations in PKA activity, with many changes in localisation and activity being stress- and isoform- specific. Under heat stress, Tpk2 and Tpk3 associate with stress-induced compartments, such as P-bodies and stress granules, with Tpk2 localisation dependent on its PrLD. Although Tpk1 localisation is unaltered, *TPK1* mRNA coalesces into cytoplasmic granules. H_2_O_2_ induces the re-localisation of Tpk1 and Tpk2, but not Tpk3, into cytosolic foci, which is linked to the nuclear accumulation of Msn2/4. As part of the response to hydrogen peroxide, glutathionylation (SG) and sulfenylation (SOH) of conserved cysteine residues in Tpk1 also occurs, specifically downregulating Tpk1 activity.

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
