# Peer review of "Yeast Protein Kinase A Isoforms: A Means of Encoding Specificity in the Response to Diverse Stress Conditions?"

_biomolecules, 2022, doi:10.3390/biom12070958_

Round 1

Reviewer 1 Report

In this review, Creamer and colleagues provide a comprehensive overview of the relationship between Protein Kinase A (PKA) and stress in yeast. The manuscript is well written and includes all the relevant point in the field. In this reviewer’s opinion, the authors could elaborate a little more in those cross sections of PKA signalling in yeast and mammalian cells. Specifically, I would suggest the following points:

1.      The authors in page 3 lane 115 cite crucial work from the Scott laboratory suggesting that dissociation of the PKA tetramer is not necessary for PKA catalytic activity. This seminal work challenged the formed dogma, however shortly after work from the Gold laboratory in PNAS (PMID: 28893983) challenged this work. The authors should comment on these discrepancies and how the yeast model could help us resolve them.

2.     While the authors discuss extensively the different modalities of PKA regulation in yeast and mammals, they completely ignore the regulatory actions of the phosphatases. For instance, the phosphatases are the final terminators of PKA signals and recently have also shown to shape the actions of PKA in the nucleus and mitochondria (please see PMID: 33742135 and PMID: 15567059). What is interesting is that yeast phosphatases are multi-lineage enzymes, contrary to kinases that all share similar ancestry. The authors should take in consideration these traits and discuss the possible roles of phosphatases in the regulation of PKA in the yeast.

3.     In a similar manner the authors do not mention the role of PDEs in shaping the PKA signals in yeast. It would be a nice addition to discuss how PDEs are involved in PKA activation and likely compartmentalisation in yeast.

Author Response

1. The authors in page 3 lane 115 cite crucial work from the Scott laboratory suggesting that dissociation of the PKA tetramer is not necessary for PKA catalytic activity. This seminal work challenged the formed dogma, however shortly after work from the Gold laboratory in PNAS (PMID: 28893983) challenged this work. The authors should comment on these discrepancies and how the yeast model could help us resolve them.

We have now included reference to work from the Gold laboratory (PMID: 28893983) and elaborated on the discrepancies between this data and work from the Scott laboratory regarding PKA activation (page 4; first paragraph).

2. While the authors discuss extensively the different modalities of PKA regulation in yeast and mammals, they completely ignore the regulatory actions of the phosphatases. For instance, the phosphatases are the final terminators of PKA signals and recently have also shown to shape the actions of PKA in the nucleus and mitochondria (please see PMID: 33742135 and PMID: 15567059). What is interesting is that yeast phosphatases are multi-lineage enzymes, contrary to kinases that all share similar ancestry. The authors should take in consideration these traits and discuss the possible roles of phosphatases in the regulation of PKA in the yeast.

We agree with the reviewer that the role of phosphatases in ultimately terminating PKA signals is important. We have now made reference to this function in the introduction (first paragraph) and Conclusions and future perspectives (Line 512).

3. In a similar manner the authors do not mention the role of PDEs in shaping the PKA signals in yeast. It would be a nice addition to discuss how PDEs are involved in PKA activation and likely compartmentalisation in yeast.

We have expanded our discussion of PDEs and now included how the role of compartmentalisation in yeast is not well understood in comparison to mammalian systems (page 3; line 99).

Reviewer 2 Report

This is a very well written article. In this review, the authors summarized the role and regulation of cAMP/PKA siganling pathway in the glucose sensing circuit in yeast. Appropriate literature search has been done and each isoform was quite well-studied. In the latter part, the effects of the PKA isoform by heat shock and oxidative stress and its regulatory action were well described. English grammer and expressions are written so that general readers can understand them well, so this review may be approved for publication immediately without modification.

Author Response

Nothing to address